# Characterization of Inflammatory Signals in BV-2 Microglia in Response to Wnt3a

**DOI:** 10.3390/biomedicines11041121

**Published:** 2023-04-07

**Authors:** Cheng Li, Ying Wu, Ming-Yue Huang, Xue-Jun Song

**Affiliations:** 1Department of Medical Neuroscience, School of Medicine, Southern University of Science and Technology, Shenzhen 518055, China; 2SUSTech Center for Pain Medicine, Southern University of Science and Technology, Shenzhen 518055, China

**Keywords:** neuroinflammation, microglial activation, Wnt3a, β-catenin

## Abstract

Activation of microglia is one of the pathological bases of neuroinflammation, which involves various diseases of the central nervous system. Inhibiting the inflammatory activation of microglia is a therapeutic approach to neuroinflammation. In this study, we report that activation of the Wnt/β-catenin signaling pathway in a model of neuroinflammation in Lipopolysaccharide (LPS)/IFN-γ-stimulated BV-2 cells can result in inhibition of production of nitric oxide (NO), interleukin-6 (IL-6), and tumor necrosis factor-α (TNF-α). Activation of the Wnt/β-catenin signaling pathway also results in inhibition of the phosphorylation of nuclear factor-κB (NF-κB) and extracellular signal-regulated kinase (ERK) in the LPS/IFN-γ-stimulated BV-2 cells. These findings indicate that activation of the Wnt/β-catenin signaling pathway can inhibit neuroinflammation through downregulating the pro-inflammatory cytokines including iNOS, TNF-α, and IL-6, and suppress NF-κB/ERK-related signaling pathways. In conclusion, this study indicates that the Wnt/β-catenin signaling activation may play an important role in neuroprotection in certain neuroinflammatory diseases.

## 1. Introduction

Neuroinflammation is a pathological process in the central nervous system (CNS) and is essential in various acute and chronic brain diseases [1,2,3]. Glial cells are the leading operators of neuroinflammation, especially for the microglia [4,5]. Microglia are the resident immunocompetent cells of CNS and play a crucial role in neurological disorders [5]. The activation of microglia produces various pro-inflammatory cytokines, including interleukin-6 (IL-6), tumor necrosis factor-α (TNF-α), and nitric oxide (NO) [6]. Inhibiting microglial activation is a potential strategy for the treatment of neurological diseases caused by neuroinflammation.

The microglial cell line BV-2 cells present a classic model of microglia-induced neuroinflammation in vitro [6,7,8]. Lipopolysaccharide (LPS) is a glycolipid that doesn’t affect the activity of BV-2 microglial cells but can accelerate the release of pro-inflammatory cytokines [9,10]. LPS can stimulate the activation of BV-2 cells and is usually used to study microglial neuroinflammation [11,12,13].

Wnt signaling is a critical regulator of neuroinflammation [14,15,16]. Wnt signaling pathways are divided into the canonical Wnt/β-catenin pathway and the non-canonical Wnt signaling pathway, including the Wnt/calcium pathway and the Wnt/planar cell polarity pathway [17]. The Wnt/β-catenin signaling pathway is crucial in maintaining neuronal homeostasis [18]. It has been reported that activation of the Wnt/β-catenin signaling pathway can inhibit pathological neuroinflammation [17,19]. In addition, the Wnt/β-catenin signaling pathway is involved in anti-tumor immunity, in which β-catenin can regulate NF-κB activation [17,20,21,22]. For instance, β-catenin can inhibit the nuclear transcription of NF-κB and can, therefore, suppress the production of cytokines, such as IL-6, IL-18, and IL-1β, in mantle cell lymphoma (MCL) [22]. However, it is still unknown whether the Wnt/β-catenin signaling can regulate the activity of CNS resident immune cells and microglia.

In this study, we utilized LPS/IFN-γ-stimulated BV-2 cells as the model of neuroinflammation. We identified the Wnt3a/β-catenin signaling as a negative modulator of microglia-produced neuroinflammation. In response to LPS/IFN-γ treatment, expression of β-catenin in BV-2 cells was significantly decreased. Incubation of exogenous recombinant protein Wnt3a (RM Wnt3a) or overexpression of Wnt3a dramatically promoted β-catenin expression in BV2 cells. Additionally, Wnt3a overexpression remarkably inhibited pro-inflammatory mediators (NO, iNOS, CD68, TNF-α, and IL-6) in LPS/IFN-γ-stimulated BV2 cells by limiting the activities of NF-κB and ERK1/2.

## 2. Methods

### 2.1. Drug Administration

Recombinant mouse Wnt3a protein (RM Wnt3a, 1324-WN) and DMSO were bought from R&D Systems (Minneapolis, MN, USA) (St. Louis, MO, USA). Drug dosages were determined based on prior studies and our preliminary investigations [23].

### 2.2. Quantitative Real-Time Polymerase Chain Reaction—RT-PCR

Using the EasyPure^®^ RNA Kit (Trans, ER101-01, Beijing, China), the manufacturer extracted the total RNA as directed. The Takara PrimeScript Master Mix (Perfect Real Time) kit was then used to create cDNA (Trans). The DyNAmo Flash SYBR Green qPCR Kit was used for the quantitative real-time polymerase chain reaction (Thermo Fisher Scientific, Waltham, MA, USA). The standard conditions were 95 °C for 7 min, followed by 40 cycles of 95 °C for 10 s, 60 °C for 30 s, 95 °C for 15 s, 60 °C for 60 s, and 95 °C for 15 s for the melt curve.

The 2-CT method was used to calculate relative mRNA levels. The relative expression of the genes of interest was then compared with the corresponding experimental control after first being normalized to the housekeeping control gene Gapdh. Primers used for expression analysis were as shown in Table 1.

### 2.3. Protein Determinations

Wnt and other signal alterations in protein levels were measured using Western blot analysis. Cell samples were lysed in ice-cold (4 °C) NP-40 or RIPA lysis buffer with a combination of a protease inhibitor, phosphatase inhibitor, and phenylmethylsulfonyl fluoride. The total protein content of the samples was equalized, and protein concentrations of the lysates were determined using the bicinchoninic acid (BCA) method (with reagents from Pierce Biotechnology). SDS-PAGE was used to separate the total proteins, and the membranes (0.2 m nitrocellulose or PVDF) were then transferred (both from Bio-Rad, Hercules, CA, USA). iNOS (1:2000; AB115819, Abcam, Cambridge, UK), anti-actin (1:1000; 4970, CST, Danvers, MA, USA), anti-Wnt3a (1:1000; 09-162, Millipore, Burlington, MA, USA), anti-β-catenin (1:1000; 610153, BD Biosciences, San Diego, CA, USA), and anti-GAPDH (1:5000; G9545, Sigma-Aldrich, St. Louis, MO, USA) were used. The membranes were then produced using PerkinElmer’s enhanced chemiluminescence reagents and secondary antibodies linked to horseradish peroxidase (R&D Systems). Quantity One 4.6.5 and the Molecular Imager (ChemiDoc XRS; Bio-Rad) were used to examine the data (Bio-Rad).

### 2.4. Immunocytochemistry

The cells were treated with 4% paraformaldehyde for 10 min, 0.1% TritonTM X-100 for 15 min, and blocked with 1%BSA for 1 h at room temperature. The cells were then incubated with the primary antibodies (anti-IBA1 (1:200; AB5076, Abcam, Cambridge, UK) and anti-CD68 (1:100; 14-0681-82, Sigma-Aldrich, St. Louis, MO, USA)) for an overnight period at 4 °C. The cells were then washed with PBS-T and incubated with the secondary antibodies (Alexa Fluor@555 anti-rat IgG (1:2000; 4417s, CST, Danvers, MA, USA), Alexa Fluor 488 (A-11055, 1:2000; Thermo Fisher Scientific, Thermo Fisher Scientific, Waltham, MA, USA), and Alexa Fluor 555 (A32816, 1:2000; Thermo Fisher Scientific, Thermo Fisher Scientific, Waltham, MA, USA)) at room temperature for 1 h.

After being washed three times for 10 min with PBS-T, DAPI (1:10,000; DUO82040, Sigma-Aldrich, St. Louis, MO, USA) was used as a counterstain on the cells. Finally, fluorescence images were captured with a fluorescence microscope (Nikon, A1, Shanghai, China), and the analysis of the fluorescence images was performed by ImageJ2.

### 2.5. ELISA

ELISA was used to measure the levels of inflammatory cytokines. We used ELISA Kits, Mouse Protein TNF-a (catalog #EK282HS-96) and Mouse Protein IL-6 (catalog #EK206/3-96) (Multi Sciences, Hangzhou, Zhejiang, China), following the manufacturers’ instructions to determine whether inflammatory factors were released in models of LPS/IFN-induced neuroinflammation. Using a microplate reader set to 450 nm, the optical density in each well was calculated following incubation with TMB (tetramethylbenzidine) substrate. The value of 630 nm was chosen as the second reading’s setting for wavelength adjustment. The concentration of inflammatory components was discovered by carrying out a standard curve following the manufacturer’s instructions.

### 2.6. Cell Culture

The BV-2 cell line (HAKATA, M011) was kept alive by feeding DMEM/F12 with 10% FBS, maintained at 37 °C, 5% CO_2_. By using the MycoAlert™ Mycoplasma Detection Kit (Lonza Group Ltd., Basel, Switzerland), BV-2 cell lines were tested for mycoplasma.

### 2.7. Nitroxide (NO) Assay

The BV-2 cells’ culture supernatant nitrate content was determined using the Griess reagent. After being plated in 96-well plates for 12 h, the BV-2 cells (1.2 × 10^5^ cells/well) were treated with RM Wnt3a (0.125, 0.25, 0.5, 1, 2, 4, 8 µM) for 12 h, and then stimulated with LPS and IFN-γ for 24 h. For the NO assay, the culture supernatant was gathered. A microplate reader was used to measure the absorbance at 540 nm, and a standard curve was used to calculate the NO concentration.

### 2.8. Cell Viability Assay

According to the manufacturer’s instructions (cas.298-93-1, Sigma-Aldrich, St. Louis, MO, USA), the 3-(4,5-dimethylthiazol-2-yl)-2,5-diphenyltetrazolium bromide (MTT) assay was used to determine the vitality of BV-2 cells under the specified conditions. Then, 96-well plates containing BV-2 cells were planted with 1 × 10^5^ cells per well. The prescribed incubation durations were followed by adding 20 µL (5 mg/mL) of MTT solution. The cells were then cultured for 4 h. A total of 200 µL of DMSO was added to each well after all the medium had been taken out to dissolve the formazan crystals that had formed. The absorbance at 570 nm was measured to obtain the results.

### 2.9. Statistics

All statistical analyses were carried out using Prism (GraphPad). All data are presented as mean ± sem. Each data point represents an individual experiment or cells. Tests of statistical difference were performed with GraphPad Prism 8 software using unpaired *t*-test (two-sided), one-way ANOVA followed by Dunnett’s multiple comparisons test. The data had normal distribution and the homogeneity of variance also met specific criteria and there was no correction for multiple testing. Sample sizes are consistent with those reported in similar studies and provide sufficient power to detect changes with the appropriate statistical analysis. For all experiments, a criterion α level was set at 0.05.

## 3. Results

### 3.1. The Expression of β-Catenin Is Inhibited in LPS/IFN-γ-Treated BV2 Cells

To test whether the Wnt3a/β-catenin signaling pathway participates in microglia-induced neuroinflammation, we used a model of BV-2 cells that exhibits morphological and functional features similar to primary microglia [1,2]. We used LPS at 100 ng/mL [24,25,26]. Through Griess reaction assays, we found that NO concentration in the supernatants of unstimulated BV-2 cells ranged from 0 to 5 µM. Furthermore, compared with separate incubation with LPS or IFN-γ, combined incubation with 100 ng/mL LPS and 20 ng/mL IFN-γ led to higher NO production in BV2 cells, with supernatant concentrations reaching roughly 20 µM (Figure 1a). IFN-γ accelerated the release of NO in a dose-dependent manner from 0 to 20 ng/mL, while 40 ng/mL IFN-γ caused a slight decrease of NO instead compared to 20 ng/mL IFN-γ (Figure 1b). The time course of NO production increased significantly within 6 h after LPS and IFN-γ exposure and peaked at 24 h (Figure 1c). Based on these results, we treated BV2 cells with 100 ng/mL LPS and 20 ng/mL IFN-γ for 24 h during the follow-up study.

It has been reported that activation of the Wnt/β-catenin signaling pathway can inhibit the pro-inflammatory activation of macrophages [4,5,6,27]. To understand whether the Wnt/β-catenin signaling pathway participates in the pro-inflammatory activation of BV-2 cells, we examined the changes of β-catenin by Western blot and immunofluorescence. Our results showed that β-catenin was significantly downregulated with exposure to LPS and IFN-γ (Figure 1d). Meanwhile, we noticed that the size of BV-2 cells was enlarged by the treatment of LPS+IFN-γ, but not the Wnt3a (Figure 1e). This finding suggests that the Wnt/β-catenin signaling may negatively regulate the activation of BV-2 cells.

### 3.2. Wnt3a Does Not Induce Cytotoxicity to BV-2 Microglial Cells

We examined the survival rate of BV-2 cells treated with RM Wnt3a to learn more about the impact of the Wnt/β-catenin signaling in BV-2 cells. The cell counts revealed that, following pro-inflammatory activation (for 24 h), there was no discernible difference in the survival rate among BV-2 cells cultured with RM Wnt3a (0.125, 0.25, 0.5, 1, 2, 4 and 8 μM) (Figure 2a). These results indicate that the Wnt/β-catenin pathway is not involved in the survival process of BV2 cells.

### 3.3. Wnt3a Activates Wnt/β-Catenin Pathway in BV-2 Cells

Western blot analysis showed that Wnt3a at 2 μM, 4 μM, and 8 μM, significantly increased the protein level of β-catenin (Figure 2b), suggesting that RM Wnt3a can induce the accumulation of β-catenin in BV-2 cells.

To confirm the role of the Wnt3a/β-catenin signaling pathway in microglia activation, we transfected BV-2 cells with pcDNA3.1-Wnt3a (Wnt3a+) to elevate the endogenous expression of Wnt3a in BV-2 cells. Western blot analysis showed that the expression of Wnt3a and β-catenin was significantly upregulated in the Wnt3a-overexpressed BV-2 cells (Figure 2c,d). Immunofluorescence staining showed that LPS treatment, but not Wnt3a overexpression, altered the morphology of BV-2 cells (Figure 2e). These results suggest that both exogenous and endogenous overexpression of Wnt3a can activate the Wnt/β-catenin signaling pathway in BV-2 cells.

### 3.4. Overexpression of Wnt3a Inhibits Pro-Inflammation Activation in LPS/IFNγ-Treated BV-2 Cells

iNOS is one of the critical pro-inflammatory cytosine that catalyzes NO production and modulates immune activation by promoting the synthesis of inflammatory factors [8,9,10]. Our Western blot analysis showed that the expression of iNOS significantly increased following combination treatment of LPS and IFNγ for 6 h and the expression peaked at 24 h (Figure 3a). Interestingly, our results showed that the increased expression of iNOS following LPS and IFNγ treatment for 24 h was significantly inhibited in the Wnt3a-overexpression BV-2 cells (Figure 3b).

Another pro-inflammation cytosine, CD68, has been reported to be strongly upregulated in microglia during neuroinflammation [11,12,13]. As shown by immunofluorescence analysis, CD68 was dramatically upregulated after exposure to LPS and IFN-γ, while such upregulation was reversed in the Wnt3a-overexpression BV-2 cells (Figure 3c). These results indicated that overexpression of Wnt3a can inhibit the pro-inflammation activation in BV-2 cells.

### 3.5. Overexpression of Wnt3a Inhibits the Expression of Pro-Inflammatory Mediators in LPS/IFNγ-Stimulated BV-2 Cells

Next, we examined the impact of activation of the Wnt/β-catenin pathway on the pro-inflammatory activation of BV-2 cells. ELISA and qRT-PCR assays were used to analyze the level of iNOS, TNF-α, and IL-6. Our results showed that LPS/IFN stimulation significantly increased the mRNA levels of iNOS, IL-6, and TNF-α in BV-2 cells, such an increased mRNA level was reversed by Wnt3a overexpression (Figure 4a–c). Meanwhile, the release of NO, IL-6, and TNF-α in the supernatants was decreased in the Wnt3a-overexpressed BV-2 cells (Figure 4d–f). These results indicate that the Wnt3a/β-catenin pathway can counteract pro-inflammation responses by inhibiting the expression of pro-inflammatory cytokines.

### 3.6. Wnt3a-Overexpression Suppresses the Activation of NF-κB and ERK1/2

NF-κB is an essential protein that causes inflammation by controlling the production of pro-inflammatory cytokines [17,18,28,29]. To clarify the mechanisms underlying the anti-inflammatory effect of the Wnt3a/β-catenin signaling, we assessed the effects of Wnt3a overexpression on LPS/IFN-γ-induced phosphorylation of NF-κB in BV-2 cells. Our results showed that the phosphorylation of NF-κB was significantly reduced (Figure 5a–c), while the baseline level of proteins was unaffected. Moreover, the expression of iNOS expression in microglia mainly depends on the activation of ERK1/2 [19]. Our results further showed that the phosphorylation of ERK1/2 was decreased in Wnt3a-overexpressed BV-2 cells (Figure 5d–f). These findings indicate that activation of the Wnt3a/β-catenin pathway can inhibit the pro-inflammatory in LPS/IFNγ-stimulated BV-2 cells by limiting the activities of NF-κB and ERK1/2.

## 4. Discussion

Microglia are resident immune cells in the CNS [30,31]. The pro-inflammatory activation of microglia is the leading cause of many brain diseases, such as autoimmune encephalitis, Parkinson’s disease, neuromyelitis optica, and Alzheimer’s disease [32,33,34]. Overactivated microglia can release pro-inflammatory cytokines, which are the major culprits in neuroinflammation. The Wnt/β-catenin signaling pathway is necessary for the survival and function of microglia [35]. Our study reveals that the activation of the Wnt3a/β-catenin signaling pathway can inhibit the release of inflammatory cytokines in BV-2 cells by suppressing the phosphorylation of NF-κB and ERK. These findings demonstrate that microglia-induced neuroinflammation may be treated by activating Wnt/β-catenin signaling pathway.

BV-2 cells are often used to investigate neuroinflammation affected by activated microglia [36,37,38]. Studies have shown that IFN-γ can induce BV-2 cell activation, which causes NO production [39,40]. Our results show that LPS can enhance IFN-γ-induced NO production, and the concentration of IFN-γ is positively correlated with NO production in a dose-dependent manner. The BV-2 cells are often used to investigate the neuroinflammation effected by activated microglia, even though they are not a complete replacement for microglia [36]. It is also possible that the BV-2 cell line may not retain the original characteristics of macroglia [36]. Further studies using primary microglia or live animals are needed.

It is well known that Wnt/β-catenin signaling plays critical roles in cell survival, proliferation, differentiation, and apoptosis [41,42]. In cancer, the Wnt/β-catenin signaling pathway participates in tumor growth by regulating inflammatory response [42,43]. There are increasing reports of the role of the Wnt/β-catenin signaling pathway in regulating inflammation [44,45]. Our study shows that, upon IFN-γ and LPS treatment, the expression of β-catenin is significantly decreased in BV-2 cells. Activated microglia produce the pro-inflammatory cytokines NO, TNF-α, and IL-6, and these cytokines are reversed in Wnt3a-overexpressed BV-2 cells. pcDNA3.1-Wnt3aBV-2 cells were used to investigate the effect of Wnt3a/β-catenin signaling pathway on inflammation. However, the secretion of Wnt3a in the spinal cord and primary microglia needs to be further examined.

NF-κB is a crucial transcription factor in inflammation and other pathological processes [46,47,48]. Studies have reported the interaction between the β-catenin and NF-kB pathways [49,50,51,52]. Here, we observed that the activation of the Wnt/β-catenin signaling pathway can inhibit LPS/IFN-γ-induced phosphorylation of NF-κB. Thus, activation of the Wnt3a/β-catenin signaling pathway may inhibit LPS/IFN-γ-induced inflammation by suppressing NF-κB activation. We know that the nuclear translocation of NF-κB is essential to inflammation. How the Wnt/β- catenin signaling pathway suppresses the nuclear translocation of NF-κB needs to be further investigated.

In addition to NF-κB, mitogen-activated protein kinase (MAPK) is essential to inflammation and microglial activation [48,53]. MAPKs, a serine/threonine protein kinase family, including p38 and ERK, are crucial in neuroinflammation [54,55]. Our results showed that activation of the Wnt3a/β-catenin pathway can inhibit ERK phosphorylation, suggesting that activation of the Wnt/β-catenin signaling pathway can inhibit inflammation through the ERK-dependent pathway in BV-2 cells. Our study demonstrates an important role of activation of the Wnt3a/ β-catenin signaling pathway in anti-inflammation in the BV-2 cell model. This cell model may be very useful for studying neuroinflammation in vitro, in conjunction with other studies in brain tissues and in vivo.

## 5. Conclusions

The Wnt/β-catenin signaling activation inhibits LPS/IFN-γ-induced neuroinflammation in BV-2 cells. In more detail, Wnt3a can inhibit the upregulation of iNOS, TNF-α, and IL-6 at the mRNA and protein levels and the secretion of NO, TNF-α, and IL-6. These processes may be related to inhibition of the ERK and NF-κB pathways. These findings suggest that targeting the Wnt/β-catenin signaling pathway can be a potential treatment approach for preventing and treating certain neuroinflammatory disorders.

## Figures and Tables

**Figure 1 biomedicines-11-01121-f001:**
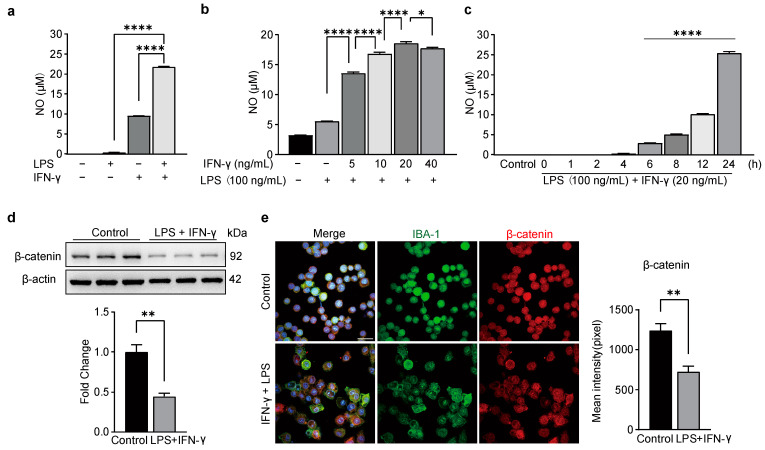
BV-2 cell activation led by LPS and IFN-γ inhibited the expression of β-catenin. A Griess reaction kit detected NO production in the supernatant (**a**–**c**). (**a**) NO production in the supernatant after incubating BV-2 cells for 24 h with LPS (100 ng/mL), IFN-γ (20 ng/mL), or both. One-way ANOVA, F (3, 8) = 52,697, **** *p* < 0.0001, *n* = 3. (**b**) The dose-dependent reaction of IFN-γ combined with LPS (100 ng/mL) induced NO production in BV-2 cells. One-way ANOVA, F (5, 30) = 1410, **** *p* < 0.0001,* *p* = 0.0191, *n* = 3. (**c**) The time course of NO production in LPS/IFN-γ-induce BV-2 cells. One-way ANOVA, F (8,18) = 3232, **** *p* < 0.0001, *n* = 3. (**d**) Western blot showing expression of β-catenin in BV-2 cells. Top representative bands. Bottom, data summary. Two-tailed unpaired *t*-test, F, DFn, Dfd (4.798, 2, 2), ** *p* = 0.0050, *n* = 3 (**e**) Left, immunofluorescence showing expression of β-catenin (red). Right, data summary. Two-tailed unpaired *t*-test, F, DFn, Dfd (1.375, 2, 2), ** *p* = 0.0089, *n* = 3.

**Figure 2 biomedicines-11-01121-f002:**
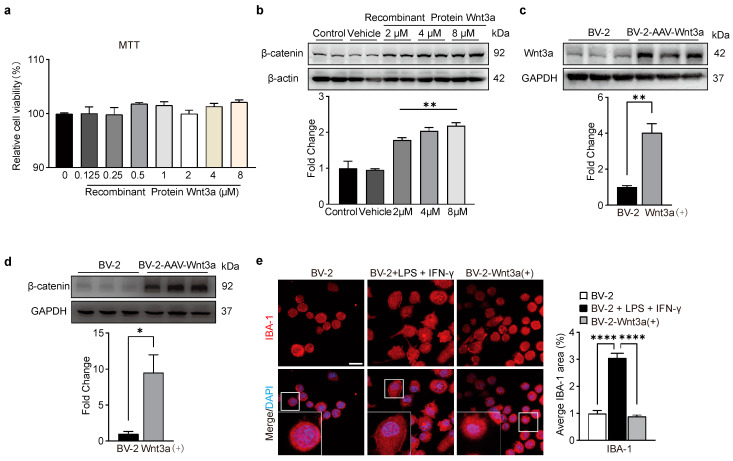
Overexpression of Wnt3a did not alter the morphology of BV-2 cells. (**a**) Treatment of BV-2 cells with RM Wnt3a at different concentrations. The cell viability was evaluated by CCK-8 assay. One-way ANOVA, F (7, 16) = 1.705, *p* > 0.05, *n* = 3. (**b**) Western blot analysis of expression of β-catenin in BV-2 cells after RM Wnt3a. Top, representative bands. Bottom, data summary. One-way ANOVA, F (4, 9) = 38.54, ** *p* < 0.005, *n* = 3. (**c**) Western blot analysis of expression of Wnt3a in pcDNA3.1-Wnt3aV-2 cells. Top, representative bands. Bottom, data summary. Two-tailed unpaired *t*-test, F, DFn, Dfd (43.85, 2, 2), ** *p* = 0.0037, *n* = 3. (**d**) Western blot analyzed the expression of β-catenin in pcDNA3.1-Wnt3aV-2 cells. Top, representative bands. Bottom, data summary. Two-tailed unpaired *t*-test, F, DFn, Dfd (51.34, 2, 2), * *p* = 0.0264, *n* = 3. (**e**) Immunofluorescence showing the morphological changes of BV2 cells following treatment of LPS plus IFN-γ, but not Wnt3a. Scale bar = 20 μm. Left, representative images. Right, data summary. One-way ANOVA, F(2, 15) = 104.1, **** *p* < 0.0001, *n* = 3.

**Figure 3 biomedicines-11-01121-f003:**
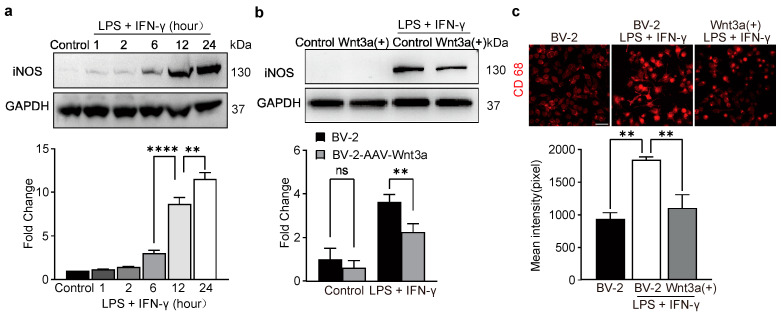
Wnt3a suppressed LPS/IFN-γ-induced upregulation of iNOS and CD68 in BV-2 cells. (**a**) Western blot showing the time course of level of iNOS. Top, representative bands. Bottom, data summary. One-way ANOVA, F (5, 12) = 107.5, **** *p* < 0.0001, ** *p* = 0.0056, *n* = 3. (**b**) Western blot showing expression of iNOS. Left, representative bands. Right, data summary. Two-way ANOVA, ** *p* = 0.0072, *n* = 3. (**c**) Immunofluorescence showing expression of CD68. Top, representative pictures. Bottom, data summary. Scale bar, 50 μm. One-way ANOVA, F (2, 6) = 13.25, **** *p* < 0.0001, *n* = 3.

**Figure 4 biomedicines-11-01121-f004:**
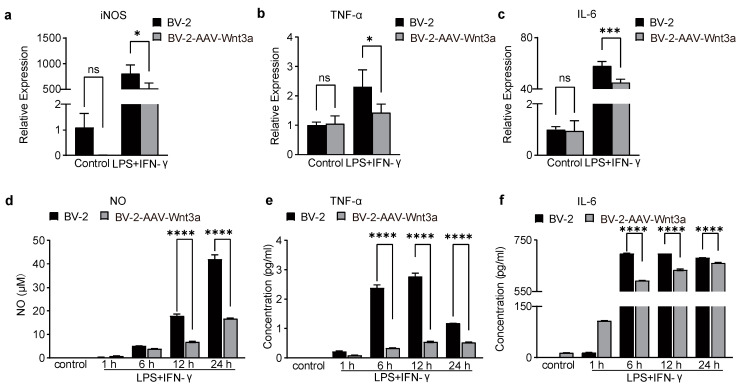
Overexpression of Wnt3a decreased NO, TNF-α, and IL-6 production in LPS and IFN-γ -stimulated BV-2 microglia cells. Overexpression of Wnt3a decreased mRNA level of (**a**) iNOS, two-way ANOVA, * *p* = 0.0139, *n* = 3. (**b**) TNF-α, two-way ANOVA, * *p* = 0.0319, *n* = 3; (**c**) IL-6, two-way ANOVA, *** *p* = 0.0002, *n* = 3. (**d**) The time course of NO production in LPS/IFN-γ-induced cells. Two-way ANOVA, **** *p* < 0.0001. *n* = 3. Wnt3a reduced the release of TNF-α. Two-way ANOVA, **** *p* < 0.0001, *n* = 3. (**e**) and IL-6 (**f**). Two-way ANOVA, **** *p* < 0.0001, *n* = 3.

**Figure 5 biomedicines-11-01121-f005:**
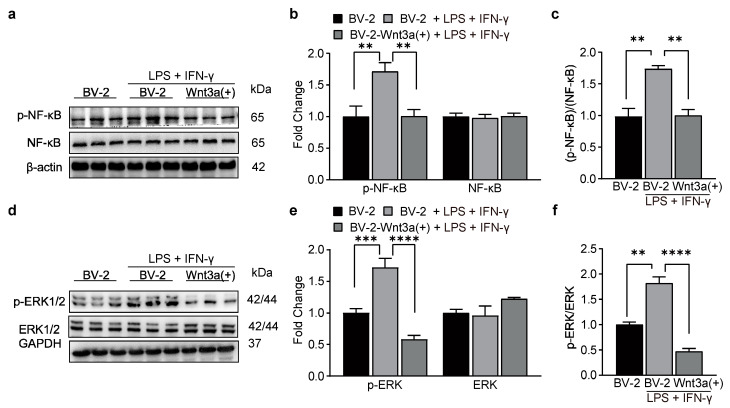
Wnt3a inhibited LPS/IFN-γ-induced activation of NF-κB and ERK. Western blot showing expression of p-NF-κB and NF-κB in BV-2 cells after LPS/IFN-γ stimulation. (**a**) Representative bands. (**b**) Data summary. Two-way ANOVA, ** *p* = 0.0010, ** *p* = 0.0012, *n* = 3. (**c**) The ratio of phosphorylation NF-κB to total NF-κB. One-way ANOVA, F (2, 6) = 20.72, ** *p* = 0.0032, ** *p* = 0.0036, *n* = 3. Western blot showing expression of p-ERK1/2 and ERK1/2 in BV-2 cells after LPS/IFN-γ stimulation. (**d**) Representative bands. (**e**) Data summary. Two-way ANOVA, *** *p* = 0.0004, **** *p* < 0.0001, *n* = 3. (**f**) The ratio of phosphorylation ERK1/2 to total ERK1/2. One-way ANOVA, F (2, 6) = 66.36, ** *p* = 0.0011, **** *p* < 0.0001, *n* = 3.

**Table 1 biomedicines-11-01121-t001:** Primers used for expression analysis.

Gene	Forward	Reverse
iNOS	GGAGTGACGGCAAACATGACT	TCGATGCACAACTGGGTGAAC
TNF-α	CAGGCGGTGCCTATGTCTC	CGATCACCCCGAAGTTCAGTAG
IL-6	CCAGAAACCGCTATGAAGTTCC	TCACCAGCATCAGTCCCAAG
β-actin	ATGGATGACGATATCGCTGC	TTCTGACCCATTCCCACCATC

## Data Availability

The data presented in this study are available on request from the corresponding author.

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
