# Peer review of "Characterization of Inflammatory Signals in BV-2 Microglia in Response to Wnt3a"

_biomedicines, 2023, doi:10.3390/biomedicines11041121_

Round 1
Reviewer 1 Report
In the manuscript by Cheng Li et al. entitled “Characterization of Inflammatory Signals in BV-2 Microglia in Response to Wnt3a” the BV-2 cells stimulated with LPS or IFN-γ were used as a model of neuroinflamation process. Authors have shown that production of nitric oxide, IL-6 and TNF-α was inhibited by the Wnt/β-catenin signaling pathway. This pathway also suppressed the phosphorylation of NF-κB and ERK related pathways.
The manuscript is well written, all sections but statistics seemed clear to me. Because there is no information how statistical tests were fit to the data and whether correction for multiple testing was applied, it is impossible to verify whether all presented results are true positive? Besides this one but major weakness of the study, all other elements are well explained and presented.
Please find below some detailed suggestions:
Introduction:
there are two dots at the end of the last paragraph in this subchapter
Methods:
please describe in detail how the statistical tests were selected - there is no information about verification of the use of parametric tests and for example there is used ANOVA, t-test (Figure 1). To use such tests the data must have normal distribution and the homogeneity of variance also has to meet specific criteria. Please write whether there was performed correction for multiple testing - without these information (requirements for a test and correction) it can not be verified whether the obtain results are true positive or false positive results.
Author Response
Dear Reviewer,
Thank you very much for your efforts and offering us this valuable opportunity to submit revision of our work and point-by-point reply.
Q1: Introduction: there are two dots at the end of the last paragraph in this subchapter.
Authors’ answer: We are sorry for many similar improper and mistakes throughout the manuscript. We have corrected them.
Q2: Methods: please describe in detail how the statistical tests were selected - there is no information about verification of the use of parametric tests and for example there is used ANOVA, t-test (Figure 1). To use such tests the data must have normal distribution and the homogeneity of variance also has to meet specific criteria. Please write whether there was performed correction for multiple testing - without these information (requirements for a test and correction) it can not be verified whether the obtain results are true positive or false positive results.
Authors’ answer: Based on the reviewer’s comments, we have added the detailed description of the statistical analyses we used in the Methods and in the Results including in each of the figure legends.
Reviewer 2 Report
In this manuscript, Li et al report that the expression of β-catenin was reduced in BV-2 cells after stimulation with LPS/IFN-γ, and activating the Wnt/β-catenin signaling pathway by overexpressing Wnt3a inhibited the inflammation response in BV-2 cells after LPS/IFN-γ treatment, as reflected by the changes of the expression of different factors such as NO, IL-6, TNF-alpha, p-NFkB and ERK. Based on these data, the authors conclude that the Wnt/β-catenin signaling pathway has the potential to act as a neuroprotective target to the 21 intervene neuroinflammatory diseases.
Previous studies have reported that decreased Wnt/β-catenin signaling drives microglial pro-inflammatory activation and stimulating this pathway could prevent the pro-inflammatory microglial activation. Therefore, the novelty of the current study is sort of compromised. With that said, the current study indeed added some details regarding the downstream signals after activating the Wnt/β-catenin pathway, which is helpful for the understanding of the effects of activating Wnt/β-catenin on anti-inflammatory response in microglia.
Some issues or concerns should be addressed:
1. I did not find the information regarding the quantification for immunocytochemistry. The relevant methods should be provided in the manuscript.
2. In Fig. 2D, no bands for β-catenin could be observed for BV-2, does it mean that normally BV-2 cells do not express β-catenin?
3. Labels of Y-axis for several figures could be improved or clarified. For example, “Relative level” was used in different figures, however, it is not clear relative to what.
4. For two-way anova analysis, it would be better including the details of the analysis, for instance, the F value, major effect or interaction effect.
5. I am confused that what statistics were used for Fig. 3a.
Author Response
Dear Reviewer,
Thank you very much for your efforts and offering us this valuable opportunity to submit revision of our work and point-by-point reply.
Q1: I did not find the information regarding the quantification for immunocytochemistry. The relevant methods should be provided in the manuscript.
Authors’ answer: We have added this information in the Methods.
Q2: In Fig. 2D, no bands for β-catenin could be observed for BV-2, does it mean that normally BV-2 cells do not express β-catenin?
Authors’ answer: Yes, the expression was relatively lower in this group compared with those with the Wnt3a treatment. We have reduced the light contrast of the bands and they look fine now.
Q3: Labels of Y-axis for several figures could be improved or clarified. For example, “Relative level” was used in different figures, however, it is not clear relative to what.
Authors’ answer: Thanks for the reviewer pointing out these important improper labels. We have replaced it with “Folder Change” in all the similar Western blot data in each of the figures.
Q4: For two-way anova analysis, it would be better including the details of the analysis, for instance, the F value, major effect or interaction effect.
Authors’ answer: As indicated by the reviewer, we have added the F values in each of the figures where the two-way ANOVA was used.
Q5: I am confused that what statistics were used for Fig. 3a.
Authors’ answer: Sorry for the mistake, we have deleted the t-test and kept the ANOVA, which was used here.
Reviewer 3 Report
A Brief Summary
The authors reported a good work demonstrating the role of Wnt/β-catenin signaling pathway in inflammation process. The authors initially tested LPS and IFN-γ to treat BV-2 cells; subsequently they evaluated the ability of Wnt/β-catenin signaling pathway to suppresses inflammation markers, nuclear factor-κB (NF-16 κB) and extracellular signal-regulated kinase (ERK).
Broad comments:
Although the study has the main limitation of a single microglia cell line, it is well structured and the experiments are conducted with good quality. I have only two major criticisms that may require minor editing before publication.
Specific comments:
1. Figure 2f. The authors write that morphology of BV-2 cells is not altered after Wnt3a overexpression, but there are significative changes in summary data. I suggest discussing the results of this figure.
2. The results of figure 5 do not show the controls (BV-2 and BV-2-AAV-Wnt3a without LPS+INF-gamma).
Minor corrections:
1. Lines 38-39: Please, clarify this sentence.
The Wnt signaling pathway that ligands from innate immune cells are critical regulators of inflammatory.
2. Line 53: Please, spell RM in the sentence, since it appears in the introduction for the first time.
Incubation of exogenous RM Wnt3a
3. Authors can enlarge discussion with broader research perspectives.
Author Response
Dear Reviewer,
Thank you very much for your efforts and offering us this valuable opportunity to submit revision of our work and point-by-point reply.
Q1: Figure 2f. The authors write that morphology of BV-2 cells is not altered after Wnt3a overexpression, but there are significative changes in summary data. I suggest discussing the results of this figure.
Authors’ answer: Sorry, we did not described clearly. It is now clarified. Immunofluorescence staining showed that LPS treatment, but not Wnt3a overexpression, altered the morphology of BV-2 cells (Figure 2e). This is noted in the Results. In addition, in our revised version, the new Figure 2e combined both Figure 2e and Figure 2f in the previous version.
Q2: The results of figure 5 do not show the controls (BV-2 and BV-2-AAV-Wnt3a without LPS+INF-gamma).
Authors’ answer: Sorry for this mistake. We mis-used the old bands from the initial experiments. We have now replaced the old bands with our new bands and data.
Minor corrections:
Q3: Lines 38-39: Please, clarify this sentence.
The Wnt signaling pathway that ligands from innate immune cells are critical regulators of inflammatory.
Authors’ answer: This sentence has now been clarified.
Q4: Line 53: Please, spell RM in the sentence, since it appears in the introduction for the first time. Incubation of exogenous RM Wnt3a.
Authors’ answer: We have spelled out the full name of RM Wnt3a when it first appears in the introduction.
Q5: Authors can enlarge discussion with broader research perspectives.
Authors’ answer: We have tried giving a little more broader research perspectives in the section of Discussion. Since our study is limited in this in vitro cell model, we hesitate to apply our results to a large scope currently.